# Dietary Protein Restriction Improves Metabolic Dysfunction in Patients with Metabolic Syndrome in a Randomized, Controlled Trial

**DOI:** 10.3390/nu14132670

**Published:** 2022-06-28

**Authors:** Rafael Ferraz-Bannitz, Rebeca A. Beraldo, A. Augusto Peluso, Morten Dall, Parizad Babaei, Rayana Cardoso Foglietti, Larissa Marfori Martins, Patricia Moreira Gomes, Julio Sergio Marchini, Vivian Marques Miguel Suen, Luiz C. Conti de Freitas, Luiz Carlos Navegantes, Marco Antônio M. Pretti, Mariana Boroni, Jonas T. Treebak, Marcelo A. Mori, Milton Cesar Foss, Maria Cristina Foss-Freitas

**Affiliations:** 1Department of Internal Medicine, Division of Endocrinology and Metabolism, Ribeirao Preto Medical School, University of Sao Paulo (USP), Ribeirao Preto 14049-900, Brazil; rebecaberaldo@yahoo.com.br (R.A.B.); larissamarfori91@gmail.com (L.M.M.); dra.patricia.moreira@gmail.com (P.M.G.); mcfoss@fmrp.usp.br (M.C.F.); 2Novo Nordisk Foundation Center for Basic Metabolic Research, Faculty of Health and Medical Sciences, University of Copenhagen, DK-1350 Copenhagen, Denmark; augusto.peluso@sund.ku.dk (A.A.P.); dall@sund.ku.dk (M.D.); parizad.babaei@sund.ku.dk (P.B.); jttreebak@sund.ku.dk (J.T.T.); 3Department of Internal Medicine, Division of Nutrology, Ribeirao Preto Medical School, University of Sao Paulo (USP), Ribeirao Preto 14049-900, Brazil; rayana.foglietti@gmail.com (R.C.F.); jsmarchini@fmrp.usp.br (J.S.M.); vmmsuen@gmail.com (V.M.M.S.); 4Department of Ophthalmology, Otolaryngology, Head and Neck Surgery, Ribeirao Preto Medical School, University of Sao Paulo (USP), Ribeirao Preto 14049-900, Brazil; lcconti@fmrp.usp.br; 5Department of Physiology, Ribeirao Preto Medical School, University of Sao Paulo (USP), Ribeirao Preto 14049-900, Brazil; navegantes@fmrp.usp.br; 6Laboratory of Bioinformatics and Computational Biology, Division of Experimental and Translational Research, Brazilian National Cancer Institute (INCA), Rio de Janeiro 20231-050, Brazil; marco-mp@hotmail.com (M.A.M.P.); mariana.boroni@inca.gov.br (M.B.); 7Program of Immunology and Tumor Biology, Division of Experimental and Translational Research, Brazilian National Cancer Institute (INCA), Rio de Janeiro 20231-050, Brazil; 8Department of Biochemistry and Tissue Biology, Obesity and Comorbidities Research Center (OCRC), Experimental Medicine Research Cluster, University of Campinas, Campinas 13083-970, Brazil; 9Department of Internal Medicine, Division of Metabolism Endocrinology & Diabetes, Caswell Diabetes Institute, University of Michigan, Ann Arbor, MI 48109, USA

**Keywords:** protein restriction, caloric restriction, type 2 diabetes, insulin resistance, metabolic syndrome, cardiovascular disease

## Abstract

Dietary restriction (DR) reduces adiposity and improves metabolism in patients with one or more symptoms of metabolic syndrome. Nonetheless, it remains elusive whether the benefits of DR in humans are mediated by calorie or nutrient restriction. This study was conducted to determine whether isocaloric dietary protein restriction is sufficient to confer the beneficial effects of dietary restriction in patients with metabolic syndrome. We performed a prospective, randomized controlled dietary intervention under constant nutritional and medical supervision. Twenty-one individuals diagnosed with metabolic syndrome were randomly assigned for caloric restriction (CR; *n* = 11, diet of 5941 ± 686 KJ per day) or isocaloric dietary protein restriction (PR; *n* = 10, diet of 8409 ± 2360 KJ per day) and followed for 27 days. Like CR, PR promoted weight loss due to a reduction in adiposity, which was associated with reductions in blood glucose, lipid levels, and blood pressure. More strikingly, both CR and PR improved insulin sensitivity by 62.3% and 93.2%, respectively, after treatment. Fecal microbiome diversity was not affected by the interventions. Adipose tissue bulk RNA-Seq data revealed minor changes elicited by the interventions. After PR, terms related to leukocyte proliferation were enriched among the upregulated genes. Protein restriction is sufficient to confer almost the same clinical outcomes as calorie restriction without the need for a reduction in calorie intake. The isocaloric characteristic of the PR intervention makes this approach a more attractive and less drastic dietary strategy in clinical settings and has more significant potential to be used as adjuvant therapy for people with metabolic syndrome.

## 1. Introduction

Metabolic syndrome constitutes a set of conditions that increase the risk of type 2 diabetes mellitus (T2D) and cardiovascular diseases (CVD), including insulin resistance, obesity, dyslipidemia, and hypertension [1,2]. Dietary restriction (i.e., 20–40% reduction in food intake) has been shown to promote longevity [3] and extend healthspan in various species [4,5], including non-human primates [6]. One explanation for the benefits of calorie restriction (CR) is the metabolic changes resulting from reducing calorie intake [7]. Sustained CR is expected to promote weight loss, contributing to improved insulin sensitivity and reduced glucose, lipid, and pro-inflammatory cytokines in the blood [8,9].

A reduction in essential macronutrient intake also accompanies CR. Thus, it remains unclear to what extent the benefits of dietary restriction are conferred by restricting specific nutrients. Some studies have demonstrated the benefits of a low-carbohydrate diet to patients with one or more conditions associated with metabolic syndrome, although the lack of control of energy intake is a potential confounder [10,11].

In modern-day society, the consumption of protein-rich diets has become more frequent and widespread due to the reduction in the costs of animal products and the westernization of feeding behavior. Protein-rich diets have been shown to increase the mortality risk associated with cardiovascular disease [12,13]. Other evidence for the deleterious effect of excess protein intake, defined by the amount in grams of protein ingested per body weight, is the association between increased circulating branched-chain amino acid (BCAA) levels and metabolic disease in mice and humans [14,15,16]. However, whether an isocaloric reduction in dietary protein intake (PR) is sufficient to mimic the effects of CR in patients with metabolic syndrome is not clear.

Furthermore, obesity and a wide range of metabolic disorders have been associated with altered gut microbiome composition, decreased microbial diversity, and reduced gene richness [17,18,19]. However, whether this pattern is altered by dietary protein restriction has not been fully elucidated.

Here we tested the hypothesis that isocaloric dietary protein restriction (PR) is sufficient to confer the beneficial effects of CR by reversing metabolic dysfunction (e.g., increased Hb1Ac levels, insulin resistance, dyslipidemia, and hypertension) in patients with metabolic syndrome. Moreover, we investigated the molecular effects of dietary restriction on adipose tissue and the impact of PR on the composition of the intestinal microbiome.

## 2. Materials and Methods

### 2.1. Experimental Design

The study was a randomized, single-center clinical trial conducted at the University of São Paulo, Brazil, between June 2017 and July 2018. The trial was registered on ensaiosclinicos.gov.br (RBR-3HKNRW) (accessed on 16 May 2015). The participants were randomized 1:1 and subjected to either CR or PR for 27 days under a strict inpatient environment, as described below. This study was approved by the Research Ethics Committee of the Clinical Hospital of Ribeirao Preto. The Helsinki Declaration Guidelines have been followed, and the reports in this article are aligned with the CONSORT standards. All participants were informed about the procedures and signed a study consent form before any study intervention.

### 2.2. Study Participants

We considered men and women aged between 25 and 60 years with a body mass index (BMI) from 28 to 40. We recruited 150 adults from all Brazilian states. The CONSORT diagram summarizing the study is provided in Appendix A. The inclusion criteria were individuals with T2D (non-insulin-dependent), hypertension, and dyslipidemia. The criteria for exclusion were the presence of health conditions such as cardiac, renal, or hepatic disease; cancer; perimenopause or irregular menstrual cycle; pregnancy; current smoking; or inability to comply with study procedures. Before admission, an initial clinical screen was carried out to ensure all individuals’ physical and psychological health.

### 2.3. Intervention Groups

Two weeks before hospital admission, all participants collected blood samples for biochemical analyses and determined body weight. Participants were also instructed to follow a diet of approximately 8368 KJ per day with a macronutrient distribution of 50% carbohydrate, 30% fat, and 20% proteins during the run-in period before admission. Examples of diets were provided as a reference for participants.

Upon admission, participants randomized to the CR group (*N* = 11) received a balanced diet (breakfast, lunch, dinner, and snacks), which had 25% less energy than the required energy consumption based on the individual’s basal determination of energy expenditure, established by indirect calorimetry. On average, the participants received a diet of 5941 ± 686 KJ per day with a macronutrient distribution of 50% carbohydrates, 30% fats, and 20% proteins.

Participants randomized to the PR group (*N* = 10) received an isocaloric diet (breakfast, lunch, dinner, and snacks) with 50% protein restriction (i.e., 0.8 g of protein/kg body weight compared to a 1.5 g of protein/kg body weight in standard Western diet). The calorie content of the diet was adjusted to match the basal energy expenditure. On average, the participants received a diet of 8409 ± 2360 KJ per day with a macronutrient distribution of 60% carbohydrates, 30% fats, and 10% proteins. Diets were prepared with similar types of food, differing only in the number of kilocalories and macronutrients. The diets were formulated with a fixed daily intake of 4 g of sodium chloride. Adherence to the diet was assessed daily after each meal by a nutritionist. All participants’ belongings were checked for additional food. Patients were hospitalized in batches of four volunteers, where two patients were randomized for CR and two for PR protocol simultaneously.

The participants were instructed not to engage in physical activity beyond the daily routine during the 27 days to avoid the well-known physical exercise effect on metabolic control.

On the first day of hospitalization, individual energy requirements were calculated after overnight fasting, based on the resting metabolic rate using indirect calorimetry (Quark RMR-Cosmed). The duration of the CR and PR protocols was 27 days. Both diets were prepared in the Research Unit of the School of Medicine of Ribeirao Preto, University of São Paulo, under a chief nutritionist’s advice.

Follow-up biochemical and anthropometric evaluations were conducted one month after the conclusion of the trial. Some participants did not return for the follow-up analyses, but we still collected data from five (45.4%) and six (60%) CR and PR protocol participants, respectively, approximately 30 days after hospital discharge.

### 2.4. Endpoints

Primary endpoints included changes in HbA_1c_ and insulin sensitivity. Secondary outcomes included changes in serum levels of total cholesterol, low-density lipoprotein cholesterol (LDL), triglycerides, fasting glucose, fasting insulin, and C-reactive protein (CRP). As exploratory endpoints, we evaluated blood pressure, body weight, body composition, fat distribution, and basal energy expenditure, and analysis of the microbiome of patients undergoing PR; analysis of the gene expression on subcutaneous adipose tissue of patients undergoing CR and PR were also analyzed. Baseline measurements were performed on the first day of hospitalization. The biochemical analyzes were performed in the central laboratory of the Clinical Hospital of Ribeirao Preto. Commercial kits were used to measure insulin (IMMULITE 2000, Malvern, PA, USA), HbA_1c_ (D-10 Hemoglobin Testing System, Bio-Rad, Hercules, CA, USA), glucose, cholesterol, triglycerides, LDL, HDL, and CRP (Wiener lab CMD 800ix2, Rosario, Argentina). Blood samples were obtained after a 12 h fast, and sample collection started at the beginning of hospitalization and continued every week until the last day of the protocol.

Energy expenditure at rest was assessed by indirect calorimetry (Quark RMR-Cosmed, Albano Laziale, Italy), where, briefly, measurement started after 20 min of rest within the outpatient setting lasting 30 min, and results were normalized by body surface area, body weight, fat mass, and free-fat mass (FFM). Insulin sensitivity was assessed by the hyperinsulinemic-euglycemic clamp described in DeFronzo et al., 1979 [20] and Ferrannini et al., 1998 [21]. Body weight was measured weekly using the same digital scale while the participant was wearing scrubs. Free-fat mass was measured every two weeks in the fasted state by electrical bioimpedance (BIA 310e Bioimpedance Analyzer-Biodynamics, Shoreline, WA, USA). Anthropometric data were collected using a tape measure and an adipometer (Lange Skinfold Caliper, Santa Cruz, CA, USA) every two weeks after 8 h of fasting.

### 2.5. Biopsy of Subcutaneous Adipose Tissue

A subcutaneous adipose tissue biopsy (40 mg) was taken after local anesthesia from 21 individuals by a trained surgeon from Ribeirao Preto Medical School-USP. Systematically, all first biopsies were collected from the right side of the abdomen (first day of each nutritional intervention), and the second biopsy from the left side of the abdomen (last day of each nutritional intervention). The adipose tissue was rapidly transported to the research laboratory. Tissue samples were rinsed in phosphate-buffered saline to remove adhering blood, and were then snap-frozen in liquid nitrogen and stored at −80 °C until completion of the study.

### 2.6. RNA-Seq

Libraries were prepared for RNA-Seq as previously described [22]. Detailed information on methods used is available in the Appendix A.

### 2.7. Single-Cell Processing and Integration

Raw matrix count of brown adipose tissue (BAT) single-nuclei RNA-Seq from Sun, et al., 2020 [23] was retrieved from the ArrayExpress database under the accession number E-MTAB-8564 (https://www.ebi.ac.uk/arrayexpress/experiments/E-MTAB-8564/ (accessed on 20 April 2021)). Data were preprocessed using the Seurat v4 package [24]. Detailed information on methods used is available in the Appendix A.

### 2.8. Cell Type Deconvolution

A normalized matrix from the BAT dataset and blood Treg from the integrated dataset was used to generate a reference matrix to estimate the different cell types in the bulk RNA-Seq data from calorie and protein restriction individuals. CIBERSORTx [25] was used to generate the reference matrix (minimum expression = 0, 5 replicates, and no sampling) and to deconvolute the bulk RNA-Seq matrix normalized to counts per million with default parameters (100 permutations). Only statistically significant results (*p*-value < 0.1) were kept. Ward. D distance was used to cluster CIBERSORTx samples, paired T-test to compare the study groups, and Pearson correlation to correlate cell types.

### 2.9. Fecal Sampling and 16S rRNA Sequencing

Samples were collected before and after diet intervention and kept at −80 °C for further procedures. Detailed information on methods used are available in the Appendix A.

### 2.10. Metagenomics Analysis

Raw DNA sequences were imported into QIIME2 (qiime2-2020.2) and quality filtered using DADA2 plugin to detect and correct sequencing errors, filtering out phiX reads, chimeric sequences, and trimming the sequences based on Phred score plots [26]. Detailed information on methods used are available in the Appendix A.

### 2.11. Estimation of Participant Group Sizes

This study’s sample size was based on feasibility for its accomplishment within the local constraints and duration of the treatment. We began with a group of 150 participants from all states of Brazil and selected 21 individuals. Due to the inclusion and exclusion criteria, how far participants lived from the Research Unit, the duration of the protocol (27 days), and the fact that patients were restricted to the hospital setting at all times during the protocol, the number of participants who could be recruited decreased. To increase treatment sensitivity, participants were randomized in a 1:1 fashion. This was a longitudinal, pairwise study limited to the hospital environment, where participants were monitored by a specialized team 24 h a day, with a strict food regimen whose composition and nature were not provided to the participants. Furthermore, using gold-standard, highly reproducible methods to assess the clinical parameters, such as the hyperinsulinemic-euglycemic clamp, the sample size was determined by pragmatic factors with 21 participants.

### 2.12. Statistical Analysis

We used ANOVA mixed-design analysis of variance to compare the effect of the diet (CR or PR, before and after intervention) and to compare the effects between diets. Tukey post hoc tests were used to interpret significant interaction effects. A paired *t*-test was performed to compare before and after treatment in the same intervention group. The results are presented as mean +/− standard deviation (SD), Δ value (initial data-final data), box blot, or dot plot. Data were analyzed without adjustment for reference values. Tests for normality were performed using the Kolmogorov-Smirnov test, and all data were normally distributed. The significance level was established at 5% in all tests. Statistical analysis and graph construction were performed using SAS (SAS Institute) version 9.0 and Graphpad Prism (Graphpad Prism 9 for Mac) version 9.0.0. Statistical testing for RNA-Seq data was performed as described in the relevant section.

## 3. Results

### 3.1. Study Overview and Characteristics of Participants

From the total of 150 patients screened, 110 were excluded based on insulin usage, age above 60 years, significant geographical distance, or refusal to participate in the study settings. Participants in the CR and PR protocol had a mean (SD) age of 49 (8.5) years and 51.6 (8.9) years, respectively. Of the participants, 63% in the CR group were women, while the PR group comprised 50% women. The mean (SD) time since the T2D diagnosis of individuals in the CR and PR groups was 9 (1.3) years and 8 (1.4) years, respectively. The mean (SD) height of the participants of the CR and PR groups was 1.64 (0.08) m and 1.65 (0.08) m, respectively (Table 1).

After the run-in period, we found no changes in fasting blood glucose, cholesterol, LDL cholesterol, triglycerides, and body weight levels (Appendix A). The 40 participants included in the study were randomly assigned to groups receiving either a balanced diet with 25% fewer calories than required based on the determination of initial basal energy expenditure (CR protocol) or an isocaloric protein restriction diet where protein intake was reduced by 50% (PR protocol). Some participants dropped out of the study due to scheduling conflicts or personal reasons. At the end of the study, 11 participants completed the CR protocol, and 10 participants completed the PR protocol (Appendix A).

### 3.2. PR and CR Promote Weight Loss and Anthropometric Changes and Increase Energy Expenditure in Patients with Metabolic Syndrome

Both interventions resulted in marked anthropometric parameters reduction in individuals with metabolic syndrome. Body weight was significantly decreased in both protocols (Figure 1A). There was a mean reduction of 6.6% in body weight in the PR group, while the CR group exhibited a mean reduction of 8%. As a consequence, BMI also decrease 6.2% (*p* = 0.001) in the PR group and 6.5% (*p* < 0.0001) in the CR group (Table 2). The changes in body weight were mainly due to a decrease in fat mass, which was reduced in both the PR and CR protocols (−9.9%, *p* = 0.0033, and −11.5%, *p* = 0.0010, respectively) (Figure 1B), whereas free-fat mass was not changed (Figure 1C). Waist circumference had a reduction of 2% (*p* = 0.0325) in the PR protocol and 5% (*p* = 0.0009) in the CR protocol (Figure 1D). Similarly, hip circumference was reduced in 2.0% (*p* = 0.0383) upon PR and in 3.5% (*p* = 0.0091) upon CR (Figure 1E). However, there was no difference in waist/hip ratio after the interventions.

We observed that energy expenditure per body surface area was decreased by 6% (*p =* 0.0311) in the CR group (Figure 2A). A decrease was also found when total energy expenditure was normalized by free-fat mass but not by body weight or fat mass (Appendix A). Moreover, mean VO₂ and VCO₂ was reduced by 7.1% (*p* = 0.0126) and 13.4% (*p* = 0.0051), respectively, after CR (Figure 2B). These changes reflected a decrease of 6.8% (*p* = 0.0296) in the respiratory exchange ratio (RER) (Figure 2C), suggesting a mild shift towards lipid metabolism. Indeed, estimation of fat oxidation showed an increase of 14% (*p* = 0.0454), while carbohydrate oxidation decreased 77.5% (*p* = 0.0452) at the end of the CR protocol (Figure 2D).

Interestingly, despite the similarities regarding changes in body weight and anthropometric parameters, PR did not significantly change energy parameters as much as CR. There were no differences in energy expenditure, VO₂, VCO₂, RER, or fat/carbohydrate oxidation in response to PR (Figure 2A–D), except for a mild decrease in energy expenditure normalized by FFM (Appendix A).

### 3.3. PR and CR Decrease Systemic Blood Pressure in Patients with Metabolic Syndrome

We investigated whether CR and PR could modify heart rate and blood pressure in individuals with metabolic syndrome. Heart rate at rest decreased from 83 ± 4 bpm to 60 ± 4 bpm with CR (*p* < 0.0001) (Figure 3A). Similarly, PR reduced heart rate from 81 ± 5 bpm to 61 ± 2 bpm (*p* < 0.0001) (Figure 3B). The CR group had a reduction of 40.9% (*p* < 0.0001) and 73.7% (*p* < 0.0001) of systolic and diastolic blood pressure, respectively (Figure 3C), while the PR group had a reduction of 37.7% (*p* < 0.0001) and 73.2% (*p* < 0.0001), respectively (Figure 3D). Remarkably, these patients went from hypertensive (CR: 159/113; PR: 162/118) to borderline normotensive (CR: 135/77; PR: 138/79) blood pressure levels.

### 3.4. PR and CR Decrease Glycemia, Circulating Lipid Levels, and Inflammation in Patients with the Metabolic Syndrome

At the end of the study, fasting blood glucose had a reduction of 52.7% (*p* = 0.0009) and 56.9% (*p* < 0.0001) under the PR and CR protocol, respectively (Figure 4A). Hemoglobin A1c (HbA_1c_) decreased in the PR and CR protocol by an average of 9.2 mmol/mol (0.85%; *p =* 0.0069) and 17.5 mmol/mol (1.6%; *p* = 0.0001), respectively (Figure 4B). PR and CR did not significantly affect serum fasting insulin, total hemoglobin, and hematocrit levels (Table 2).

In addition to changes in glucose levels, there were observed reductions in total cholesterol (35.4% in the PR protocol, *p =* 0.0047; and 34.4% in the CR protocol, *p* = 0.0002) and low-density lipoprotein (LDL) cholesterol (38.6% in the PR protocol *p =* 0.0026; and 36.9% in the CR protocol, *p* < 0.0001) after the diets (Figure 4C,D). Triglyceride levels also decreased by 39.5% (*p =* 0.0097) in the PR protocol and 49.9% (*p* = 0.0004) in the CR protocol (Figure 4E). The concentration of free fatty acids was also reduced in both protocols, PR (22.2%, *p =* 0.04) and CR (28.5%, *p =* 0.02), similarly to glycerol levels (31.6%, *p =* 0.03 in the PR; and 37.4%, *p =* 0.004 in the CR protocol protocol), indicating reduced lipolytic rate (Table 2).

The inflammatory marker C-reactive protein was decreased by 69.4% (*p =* 0.0041) and 57% (*p* = 0.0002) in the PR and CR protocol, respectively (Table 2). Creatine phosphokinase (CPK), a marker of muscle damage, showed a reduction in PR and CR of 51.1% (*p =* 0.03) and 47.7% (*p =* 0.04), respectively. In contrast, there were no changes in markers of kidney damage measured by urinary creatinine and urea levels (Table 2).

### 3.5. PR and CR Improves Insulin Sensitivity in Patients with Metabolic Syndrome

Data from the hyperinsulinemic-euglycemic clamp showed that at the end of the CR protocol, patients exhibited a 62.3% increase in insulin sensitivity (*p* < 0.0001), as determined by the rise in glucose infusion rate (GIR) (Figure 5A). Figure 5B shows the stability of blood glucose levels during the clamp, with a reduction of the area under the curve after the CR protocol (*p* < 0.0001), reinforcing the improvement in insulin sensitivity (Figure 5C). Furthermore, there were differences in the homeostasis model of beta cell function (HOMA-Beta) and the homeostasis model of insulin resistance (HOMA-IR), with an increase of 473% (*p* = 0.0056) and a reduction of 61.1% (*p* = 0.0057), respectively (Figure 5D,E).

Insulin sensitivity was also increased at the end of the PR protocol, as shown by the 93.2% (*p* < 0.0001) increase in GIR (Figure 5F). Figure 5G shows the stability of blood glucose levels during the clamp with a reduction of the area under the curve after the PR protocol (*p* < 0.0001) (Figure 5H), similar to CR. Moreover, HOMA-Beta increased by 300% (*p* = 0.0199) (Figure 5I) and HOMA-IR reduced 56.6% (*p* = 0.046) after PR (Figure 5J). A careful analysis of HOMA-β and HOMA-IR, excluding two outlier samples from both CR and PR protocols, confirmed that CR shows improved insulin sensitivity as measured by HOMA-β (*p* = 0.0132) and HOMA-IR (*p* = 0.0090). PR confirmed its improvement in beta cell function as measured by HOMA-β (*p* = 0.0025); however, HOMA-IR showed loss of significant difference (*p* = 0.0835) after excluding the two PR outlier samples (Appendix A).

### 3.6. Metabolic Improvement Is Maintained during 1-Month Follow-Up

The follow-up evaluation 30 days after hospital discharge showed maintenance of metabolic parameters obtained after both interventions. There were no significant changes in fasting glucose, cholesterol, LDL cholesterol, triglycerides, or body weight (Appendix A) between the moment of hospital discharge and the end of the 30 days follow-up evaluation. These results suggest that the diets’ effects can be sustained for at least one month outside the hospital setting.

### 3.7. PR Has No Short-Term Effects on Gut Microbiota Composition

The fecal sample analysis from seven individuals before and after the PR protocol showed no significant differences in microbiota diversity in terms of species number or richness (alpha-diversity). Beta diversity analysis could not point out any specific clustering of the samples. Differential abundance tests for composition comparing before versus after PR samples detected no significant changes. However, some species of bacteria, especially Firmicutes, showed dynamism concerning their abundance, reaching a quantity change of up to four times (fold change) (Appendix A).

### 3.8. PR and CR Short-Term Dietary Restriction Effects on SAT

The genes altered by PR or CR are shown in volcano plots in Figure 6A,B and heatmaps in Figure 6C,D, respectively. In total, approximately 24,000 genes were evaluated. We did not observe significant changes (considering false discovery rate < 0.05) in the adipose tissue transcriptome in patients undergoing either PR or CR. However, an exploratory analysis using the fold change (*p*-value < 0.05) showed that 993 genes were altered after PR, 1444 genes had alterations after CR, and 72 genes shared alterations in both nutritional interventions (Figure 6E). PR had 469 upregulated genes and 524 downregulated genes, while CR had 1087 upregulated genes and 357 downregulated genes; 15 upregulated genes and 11 downregulated genes shared alterations between the two interventions (Figure 6F). However, 46 genes had opposite directions between PR and CR, suggesting that the similarities between the PR and the CR effect could be random. Multiple pathways were enriched after PR (FDR < 0.05); however, we did not see significant changes after CR (FDR < 0.05). After PR, terms enriched among the upregulated genes included “B cell receptor signaling pathway”, “B cell proliferation”, “lymphocyte proliferation”, “mononuclear cell proliferation”, and “leukocyte proliferation”. Terms enriched among the downregulated genes included “cell migration involved in heart development”, “peptidyl-lysine oxidation”, and “oxygen homeostasis” (Figure 6G).

To investigate if the changes in gene expression were related to alterations in tissue cellularity, we performed a deconvolution analysis of our data and compared it to single-cell or single-nucleus RNA-Seq data from publicly available studies that used adipose tissue and blood samples. Our analyses showed no significant cell composition differences after PR or CR (Appendix A).

## 4. Discussion

Here we show that patients with metabolic syndrome undergoing short-term (i.e., 27 days) PR or CR interventions benefit from a wide range of clinical improvements, including reduced adiposity, normalization of blood pressure, improved insulin sensitivity, decreased glucose and lipid levels, and reduction of systemic inflammation. Notably, the impact of PR and CR appears to be persistent for at least one month after hospital discharge. Based on previous studies in humans and animal models, the effects of CR were expected [27,28,29], but the data from the PR intervention observed in this study, particularly in humans, are novel. We found that restriction of calories is not necessary to improve several metabolic parameters. Instead, PR is sufficient to confer almost the same clinical outcomes as CR without the need for a reduction in calorie intake.

Despite the differences in calorie intake between the PR and CR diets, both protocols led to weight loss, mainly due to decreased fat mass and preservation of FFM. No changes in urinary creatinine or urea excretion after the regimens confirmed the absence of protein catabolism. Maintenance of FFM is crucial to these patients, given that muscle mass is important for adequate glucose control and preserved motor functions in individuals undergoing dietary interventions [30,31].

As other authors have reported [10,11], we observed that basal energy expenditure and heart rate at resting state of participants undergoing CR is reduced, suggesting a lower metabolic rate. Our data add to these previous studies that used a long-term CR exposure (i.e., between six months and two years) by demonstrating that a short (27 days) dietary restriction regimen in a controlled environment can decrease energy expenditure in patients with metabolic syndrome. It is worth pointing out that another difference between these studies and ours is that the individuals in our study were overweight, diabetic, dyslipidemic, and hypertensive.

Interestingly, despite the alterations in body weight and adiposity, we did not see changes in basal energy expenditure in individuals subjected to PR, except for a mild decrease in energy expenditure when normalized by FFM. This contrasts with animal models of dietary methionine restriction studies, where increased respiration rates and energy expenditure were associated with decreased adiposity and reduced body weight [32,33]. Given that neither energy intake nor total energy expenditure is affected by PR in our study and given that the individuals in the PR group had an isocaloric diet personalized to their basal energy expenditure, it is not clear how these individuals lost weight. A longitudinal study with a larger number of participants is needed to comprehensively assess energy balance in individuals undergoing PR.

Previous studies have shown that decreased BMI, adiposity, and waist and hip circumference are linked to a reduced risk of CVD and reduced all-cause mortality in subjects exposed to either CR or PR [34,35,36]. Our data showed that metabolic syndrome patients undergoing PR or CR for 27 days have their blood pressure normalized and a reduction in circulating lipid levels, suggesting that the diets evaluated can mitigate CVD risk. It is important to mention that the daily sodium chloride intake fixed was at 4 g for all patients, and the reduction in blood pressure levels was associated with a reduction in adiposity. While it is known that CR reduces risk factors associated with cardiovascular disease onset (e.g., elevated blood pressure and heart rate [9]), our data show that dietary protein restriction can be a feasible strategy to control blood pressure.

Since we tested the intervention in patients with increased cardiovascular disease due to metabolic syndrome to avoid increasing fat consumption, patients had to increase their carbohydrate intake to reduce protein and keep the energy content of the diet unaltered in the PR intervention. The carbohydrate that was increased in the diet was distributed over several meals during the day. Despite this increase in carbohydrates, PR led to reductions in blood glucose and HbA_1c_ levels, and improved insulin sensitivity and beta-cell function (as assessed by HOMA-Beta). These data suggest that dietary proteins may play an important role in glucose homeostasis in patients with metabolic syndrome. This is not entirely surprising, given that essential amino acids are known to trigger nutrient-sensing pathways controlling insulin action and glucose production [37,38].

Interestingly, when comparing the amplitude of PR’s effects with the effects of CR in individuals with metabolic syndrome, we found that CR is more effective in reducing BMI and the levels of HbA_1c_, triglycerides, and glycerol. In contrast, PR is more effective in lowering CPK and urinary urea levels, as expected, with reduced protein intake. Remarkably, PR was also more effective in promoting insulin sensitivity as measured by GIR. PR has been shown to improve insulin sensitivity as measured by HOMA-IR. With a more detailed analysis, removing two PR outlier samples, we noticed a loss of significant changes (*p* = 0.0835). However, we prefer to clarify our findings on the assessment of insulin sensitivity based on the gold standard test evaluated by the hyperglycemic-euglycemic clamp, which proved the insulin improvement after PR treatment.

Together, these results demonstrate that proteins act as key dietary components for body mass and cardiometabolic function. Indeed, protein intake may serve as a true food intake signal since humans can endogenously produce glucose and fatty acids but not certain amino acids. From this perspective, considering that essential amino acids or derived metabolites signal the level of food intake and elicit a response proportional to this level, it would be expected that low protein intake results in a metabolic response similar to caloric restriction. In nature, reduced protein intake is almost always accompanied by reduced food intake, and species are likely to have evolved to identify protein availability with the diet’s metabolic response.

Compliance with a CR intervention is expected to be low in the long term, considering that feeding behavior is linked to social interactions and provides a sensation of happiness. For obese individuals, this challenge may be even harder to overcome. Our study proposes an alternative diet to help patients with metabolic syndrome lose weight and control blood glucose, lipid levels, and blood pressure without a calorie restriction regimen. Although the most adequate PR diet needs to be defined individually, examples of general PR diet programs, like veganism, have already been shown to provide humans metabolic benefits [39]. Hence, our study paves the way for future investigation to better understand PR diets’ benefits as adjuvants to treat cardiometabolic diseases.

Our results also provide an interesting discovery that individuals with metabolic syndrome undergoing the isocaloric PR diet, despite having improved many important metabolic parameters, do not show significant changes in their gut microbiome composition. This suggests that the metabolic benefits of PR are not associated with changes in the gut microbiota. However, previous studies have reported that changes in food intake modify the composition of the microbiome, such as after adding individual nutrients [40] or after a change from plant-based to animal-based diets [41]. Additionally, significant shifts in the microbiome composition were observed after immigration of individuals from the east to the west, mainly due to adaptation to the Western diet [42]. Some bacterial clusters were found to change significantly, especially on animal-based diets [41]. The maintenance of the gut microbiota composition in individuals from the PR group may be due to the significant heterogeneity of the population analyzed in this study (i.e., individuals from different states of Brazil). In addition, it is important to mention that the diet provided to the individuals in the PR group in our study was mainly plant-based, with a small amount of meat. Our data thus agree with earlier findings, which suggest that an animal-based diet has a greater impact than a plant-based diet on the gut microbiota [41]. However, future studies of the microbiome with a larger cohort of individuals undergoing an isocaloric PR diet are necessary to validate our findings. Nonetheless, to the extent that the gut microbiome may serve as a sensitive readout to detect how environmental changes impact human physiology, one could suggest that no changes in gut microbiome composition comparing individuals before and after PR reveals that hospitalization is not a major influence, under these settings, over the individuals’ ability to respond to the diet.

Adipose tissue contains multiple cell types, including adipocytes, preadipocytes, endothelial cells, stromal cells, and several immune cell subtypes [43]. This study evaluated the molecular effects of PR and CR using bulk RNA-Seq of subcutaneous adipose tissue of individuals undergoing nutritional interventions. Although the gene expression changes were not robust, there was significant enrichment of terms related to leukocytes among the genes affected by PR. This prompted us to investigate if there were changes in immune cell numbers after the interventions. According to the deconvolution analysis of the bulk RNA-Seq data, there were no significant changes in adipose tissue cellularity after PR or CR. This suggests one of the two possibilities: (i) our method was not sensitive enough to detect subtle changes in cell composition, or (ii) the gene expression changes reflect alterations in cell state and not cellular composition. Further studies are necessary to better understand this effect and determine the mechanism involved.

Our study has some limitations that we must address. Due to the long duration of treatment protocols (27 days) and constant monitoring in the hospital environment, our study sample was limited to 21 individuals. Despite the important metabolic improvements observed in individuals undergoing PR, we did not observe changes in the microbiota composition. This outcome can be associated with the small number of samples evaluated and/or time of treatment. We also did not have enough samples from patients undergoing CR to assess the microbiota. Finally, the use of Bulk RNA-seq as a methodology to evaluate gene expression in adipose tissue in this study was not sensitive enough to find possible molecular alterations in the analyzed samples; thus, the use of more adequate methodologies such as single-cell RNA-seq could clarify our findings.

## 5. Conclusions

In conclusion, we found that PR mimicked CR in regard to the amelioration of symptoms of metabolic syndrome, suggesting that protein is a primary driver underlying the beneficial effects of food restriction. Our data suggest that PR may be a relatively simple dietary intervention to help control glycemia, lipid levels, body weight, and blood pressure in individuals with metabolic syndrome.

## Figures and Tables

**Figure 1 nutrients-14-02670-f001:**
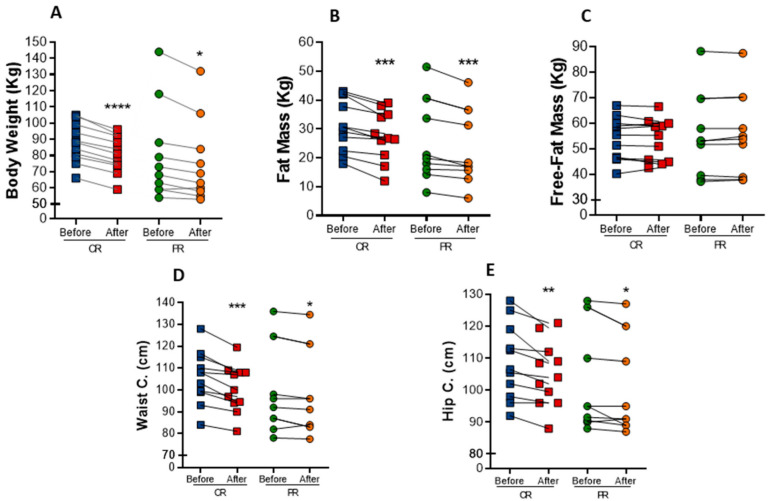
Anthropometric changes after caloric restriction (CR) or protein restriction (PR). (**A**) Body weight (**** *p =* < 0.0001 and * *p* = 0.0176). (**B**) Fat mass comparing before and after caloric or protein restriction (*** *p* = 0.0010 and ** *p* = 0.0033). (**C**) Free-fat mass comparing before and after caloric or protein restriction ( *p* = 0.7929 and *p* = 0.6464). (**D**) Waist circumference comparing before and after caloric or protein restriction (*** *p* = 0.0009 and * *p* = 0.0325). (**E**) Hip circumference comparing before and after caloric or protein restriction (** *p* = 0.0091 and * *p* = 0.0383). Data are presented in dot-plot format. ANOVA mixed-design analysis of variance was performed.

**Figure 2 nutrients-14-02670-f002:**
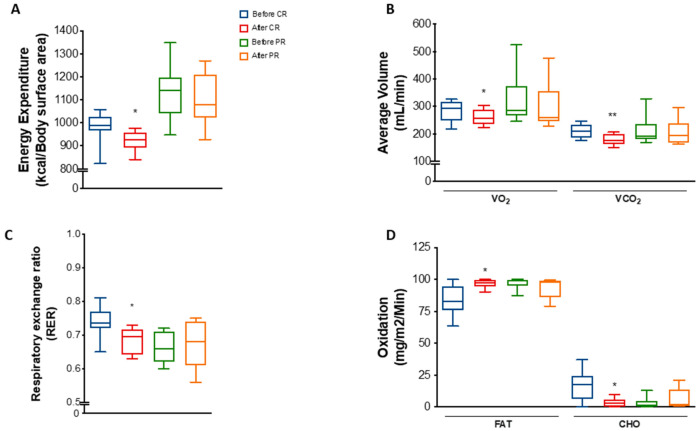
Energy expenditure and substrate utilization after caloric restriction (CR) or protein restriction (PR). Data obtained using indirect calorimetry. (**A**) Energy expenditure before and after CR (* *p* = 0.0311) and before and after PR (^ns^ *p* = 0.7568). (**B**) Average volume of VO_2_ and VCO_2_ before and after CR (* *p* = 0.0126 and ** *p* = 0.0051) and before and after PR (^ns^ *p* = 0.2736 and ^ns^ *p* = 0.5507). (**C**) RER (respiratory exchange ratio) before and after CR (* *p* = 0.0296) and before and after PR (^ns^ *p* = 0.8571). (**D**) Lipid (FAT) and carbohydrate (CHO) oxidation before and after CR (* *p* = 0.0454 and * *p* = 0.0452) and before and after PR ( *p* = 0.4281 and *p* = 0.4270). Data are shown in box plot format. ANOVA mixed-design analysis of variance was performed.

**Figure 3 nutrients-14-02670-f003:**
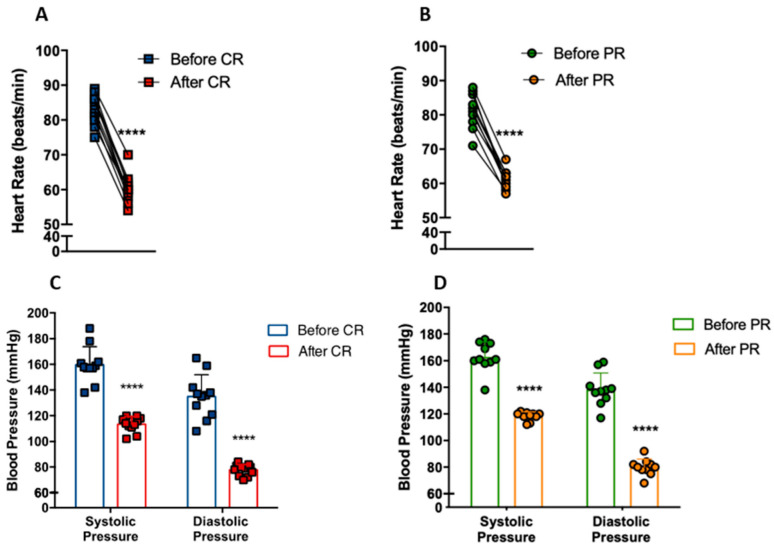
Reduction of cardiovascular disease risk factors after caloric restriction (CR) or protein restriction (PR). (**A**) Heart rate at rest before and after caloric restriction (**** *p* = < 0.0001). (**B**) Heart rate at rest before and after protein restriction (**** *p* = < 0.0001). (**C**) Blood pressure before and after caloric restriction (**** *p* = < 0.0001). (**D**) Blood pressure before and after protein restriction (**** *p* = < 0.0001). Data are presented in dot-plot format. Paired *t*-test was performed.

**Figure 4 nutrients-14-02670-f004:**
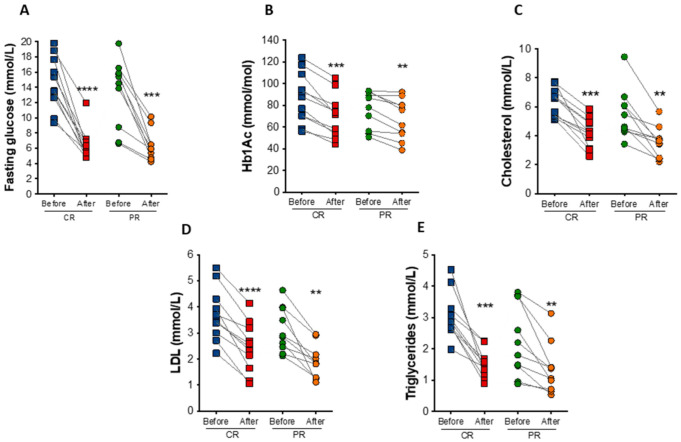
Biochemical changes after caloric restriction (CR) or protein restriction (PR). (**A**) Fasting glucose levels (**** *p* < 0.0001 and *** *p* = 0.0009). (**B**) Hemoglobin A1c (HbA1c) levels (*** *p* = 0.0001 and ** *p* = 0.0069). (**C**) Total cholesterol levels (*** *p* = 0.0002 and ** *p* = 0.0047). (**D**) low-density lipoprotein (LDL) cholesterol levels (**** *p =* <0.0001 and ** *p* = 0.0026). (**E**) Triglycerides levels (*** *p* = 0.0004 and ** *p* = 0.0097). Data are presented in dot-plot format. ANOVA mixed-design analysis of variance was performed.

**Figure 5 nutrients-14-02670-f005:**
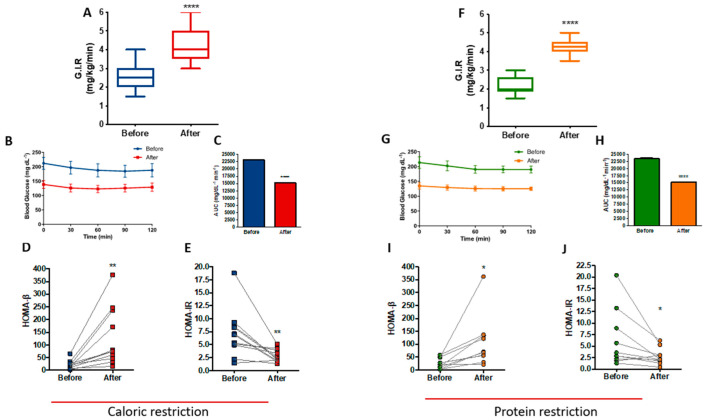
Improved insulin sensitivity after caloric restriction (CR) or protein restriction (PR). Results obtained using hyperinsulinemic-euglycemic clamp or HOMA-β and HOMA-IR. (**A**) Glucose infusion rate (GIR) comparing before and after caloric restriction (**** *p* = < 0.0001). (**B**) Glycemic curve during the clamp. (**C**) Area under the curve of B (**** *p* = < 0.0001). (**D**) HOMA-β comparing before and after caloric restriction (** *p* = 0.0056). (**E**) HOMA-IR comparing before and after caloric restriction (** *p* = 0.0057). (**F**) GIR comparing before and after protein restriction (**** *p* = < 0.0001). (**G**) Glycemic curve during the clamp. (**H**) Area under the curve of G (**** *p* = < 0.0001). (**I**) HOMA-β comparing before and after protein restriction (* *p* = 0.0199). (**J**) HOMA-IR comparing before and after protein restriction (* *p* = 0.0460). Data are shown in box plot and dot plot format. Paired *t*-test was performed.

**Figure 6 nutrients-14-02670-f006:**
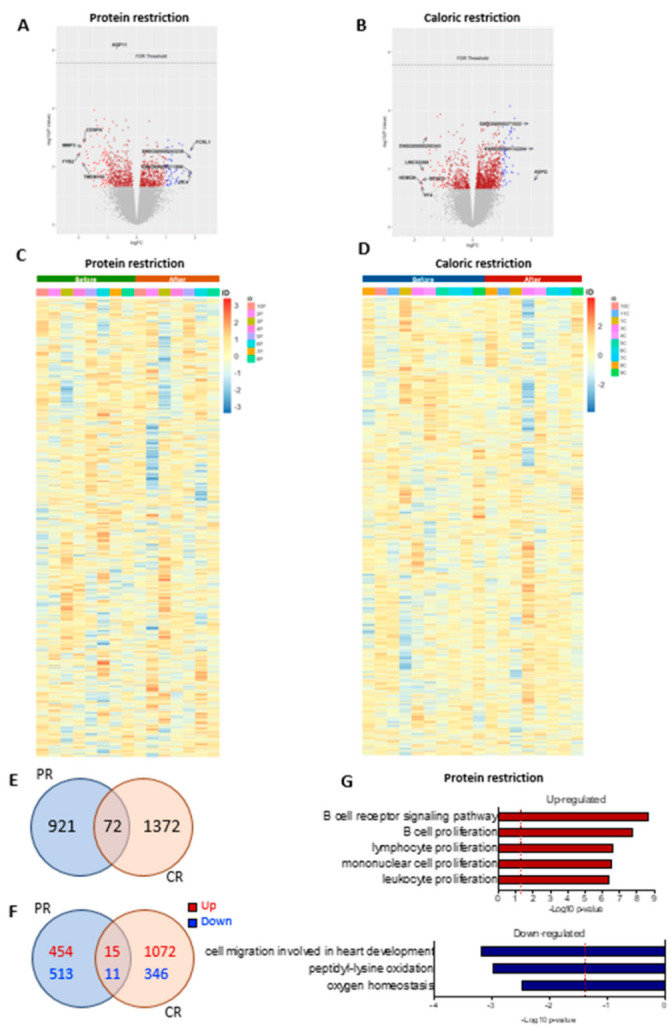
Differential gene expression after protein restriction (PR) or caloric restriction (CR). Volcano plots (log2 fold change against −log10 of the *p*-value) showing distribution of differentially expressed of the ~24,000 transcripts (**A**) before and after PR, and (**B**) before and after CR. Heatmap representation of gene expression (upregulation in red and downregulation in blue) of the individual samples comparing before and after PR (*n* = 8) (**C**), and before and after CR (*n* = 10) (**D**). Venn diagrams of adipose tissue and overlapping genes regulated by PR or CR: (**E**) total genes, (**F**) up- and downregulated genes; *p*-value < 0.05. GO terms enriched in genes upregulated and downregulated in adipose tissue after PR (**G**).

**Table 1 nutrients-14-02670-t001:** General characteristics of the subjects and primary and secondary outcomes by CR and PR.

Characteristic	CR (*n* = 11)	PR (*n* = 10)	Difference between CR and PR Effect	*p*-Value for Difference between CR and PR Effect
Age, mean (SD), y	49 (8.5)	51.6 (8.9)	2.6	0.52
Female, no. (%)	7 (63)	5 (50)	2	-
T2DM diagnostic, mean (SD), y	9 (1.3)	8 (1.4)	1	0.24
Height, mean (SD), m	1.64 (0.08)	1.65 (0.08)	−0.01	0.96
**Primary Outcomes**				
Δ Fasting glucose, mean (SD), mmol/L	−8.3 (3.0)	−7.0 (3.6)	1.3	0.20
Δ Hemoglobin A_1_c, mean (SD), mmol/mol	−17.5 (7.9)	−9.2 (6.5)	8.3	0.02
Δ Hemoglobin A_1_c, mean (SD), %	−1.6 (0.7)	−0.85 (0.6)	0.75	0.02
Δ Weight loss, mean (SD), Kg	−6.9 (1.9)	−5.4 (3.6)	1.5	0.12
Weight loss, mean (SD), %	8 (2.1)	6.6 (4.5)	1.4	0.29
**Secondary Outcomes**				
Δ Cholesterol, mmol/L	−2.2 (1.0)	−1.9 (1.3)	0.3	0.33
Δ LDL cholesterol, mmol/L	−1.4 (0.6)	−1.2 (0.7)	0.2	0.29
Δ HDL cholesterol, mmol/L	−0.2 (0.2)	−0.2 (0.2)	0	0.48
Δ Triglycerides, mmol/L	−1.5 (0.8)	−0.9 (0.7)	0.6	0.03
Δ BP Systolic, mmHg	−46.4 (25)	−44.6 (24)	1.8	0.38
Δ BP Diastolic, mmHg	−57.3 (31)	−58.5 (31)	−1.2	0.42

**Table 2 nutrients-14-02670-t002:** Changes in clinical parameters before and after CR and PR.

Variable	Caloric Restriction (*n* = 11)			Protein Restriction (*n* = 10)			CR vs. PR	*p*, CR vs. PR
	Before	After	*p*	Before	After	*p*	Effect	Effect
BMI, mean (SD), kg/m^2^	32.0 (5.4)	29.9 (5.1)	<0.0001	29.1 (8.2)	27.3 (7.0)	0.010	0.3	0.05
Insulin, mean (SD), pmol/L	77.6 (53.3)	73.3 (37.8)	0.96	71.4 (57.8)	71.9 (37.3)	0.99	3.8	0.37
CPK, mean (SD), µkat/L	1.9 (1.2)	1.0 (0.4)	0.04	2.1 (1.2)	1.0 (0.3)	0.03	−0.2	0.0001
CRP, mg/L	9.2 (5.5)	3.9 (3.8)	0.0002	8.5 (5.3)	2.6 (2.3)	0.0041	−0.6	0.48
Free fatty acids, mean (SD) *	0.7 (0.2)	0.5 (0.1)	0.02	0.9 (0.2)	0.7 (0.2)	0.04	0	0.48
Glycerol, mean (SD) *	3.0 (1.3)	1.9 (0.5)	0.004	1.7 (0.9)	1.1 (0.6)	0.03	0.5	0.0001
Urinary creatinine, mean (SD) *	6.0 (2.3)	6.4 (2.3)	0.37	4.9 (2.0)	5.8 (5.2)	0.91	−0.5	0.01
Urinary urea, mean (SD) *	345.7 (150.6)	347 (144.5)	0.99	247 (85.1)	207.9 (99.6)	0.62	−42.0	<0.0001
Hb, mean (SD), g/L	133 (10)	132 (9)	0.26	135 (11)	134 (8)	0.18	0	0.50
Ht, mean (SD), %	41.0 (3.5)	40.4 (3.1)	0.18	41.4 (4.7)	41.1 (4.0)	0.09	0.3	<0.0001

(* mmol/L); *p*: *p*-value.

## Data Availability

The datasets collected and analyzed during the current study are available from the corresponding author on reasonable request.

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
