# Peer review of "Dietary Protein Restriction Improves Metabolic Dysfunction in Patients with Metabolic Syndrome in a Randomized, Controlled Trial"

_nutrients, 2022, doi:10.3390/nu14132670_

Round 1
Reviewer 1 Report
Manuscript Number: nutrients-1751807
Dietary Protein Restriction Reverses Metabolic Dysfunction in Patients with Metabolic Syndrome in a Randomized, Controlled Trial
The authors of the original data review manuscript titled “Dietary Protein Restriction Reverses Metabolic Dysfunction in Patients with Metabolic Syndrome in a Randomized, Controlled Trial” confronts an important question regarding the effects of PR versus CR in humans. While many studies in rodents have identified the effects of PR on metabolic health and lifespan in both lean and obese conditions, this study highlights the improvements made during PR. The PR dietary interventions are largely supported by feeling less hungry as observed during CR, though CR is established as the gold standard to improve health. The data from this manuscript aligns with many studies to improve the translational aspects of PR on health as well in specific tissues, namely adipose tissue and the gut.
Overall, this manuscript is well written, providing new data regarding the effects of PR during diabetes with obesity.
Minor Concerns:
1. The title uses “reverses” in the context as to cure metabolic dysfunction, while the effects of
metabolic dysfunction are controlled from the stated metabolic endpoints only during dietary
interventions. The authors may want to consider something along the lines
- Dietary Protein Restriction Improves Metabolic Health in 2 Patients with Metabolic Syndrome in a Randomized, Controlled Trial.
- Dietary Protein Restriction Improves Metabolic Endpoints in 2 Patients with Metabolic Syndrome in a Randomized, Controlled Trial.
- Measures of Metabolic Health are Improved during Dietary Protein Restriction……
- Dietary Protein Restriction Improves Metabolic Endpoints in 2 Patients with Metabolic Syndrome in a Randomized, Controlled Trial.
This reviewer makes this suggestion due to Suppl Fig 4, where all improvements return to basal line.
2. Abstract conclusion statement: The conclusion statement presents a pretentious undertone PR compared to CR “the isocaloric characteristic of the PR intervention makes this approach a more attractive and less drastic dietary strategy in clinical settings and has more significant potential to be used as adjuvant therapy for people with metabolic syndrome”.
Major Concerns
3. Figure 2: The authors used an ANOVA for the statistical reference. Given that the body weights are different after the intervention the appropriate stat reference should be an ANCOVA, this will consider the variations and changes in BW before and after the dietary interventions for each subject. Though Figure 3 in the supplemental seems to be more accountable for changes seen during the intervention by plotting value -X by corresponding BW or changes in energy ---see publication Tim Hollstein, Alessio Basolo, Takafumi Ando, Jonathan Krakoff, Paolo Piaggi, Reduced adaptive thermogenesis during acute protein-imbalanced overfeeding is a metabolic hallmark of the human thrifty phenotype, The American Journal of Clinical Nutrition, Volume 114, Issue 4, October 2021, Pages 1396–1407, https://doi.org/10.1093/ajcn/nqab209
4. Gut microbiota. The authors state that there is no changes in the microbiome during short-term PR, yet make no mention of the CR diets or having what is consider NP (normal protein). Please discuss when interpreting your results
- Can the authors expound on the differences of CR and PR… see
1. Martin A, Ecklu-Mensah G, Ha CWY, Hendrick G, Layman DK, Gilbert J, Devkota S. Gut microbiota mediate the FGF21 adaptive stress response to chronic dietary protein-restriction in mice. Nat Commun. 2021 Jun 22;12(1):3838. doi: 10.1038/s41467-021-24074-z. PMID: 34158480; PMCID: PMC8219803.
2. Dong TS, Luu K, Lagishetty V, Sedighian F, Woo SL, Dreskin BW, Katzka W, Chang C, Zhou Y, Arias-Jayo N, Yang J, Ahdoot A, Li Z, Pisegna JR, Jacobs JP. A High Protein Calorie Restriction Diet Alters the Gut Microbiome in Obesity. Nutrients. 2020 Oct 21;12(10):3221. doi: 10.3390/nu12103221. PMID: 33096810; PMCID: PMC7590138.
3. Chen X, Song P, Fan P, He T, Jacobs D, Levesque CL, Johnston LJ, Ji L, Ma N, Chen Y, Zhang J, Zhao J, Ma X. Moderate Dietary Protein Restriction Optimized Gut Microbiota and Mucosal Barrier in Growing Pig Model. Front Cell Infect Microbiol. 2018 Jul 18;8:246. doi: 10.3389/fcimb.2018.00246. PMID: 30073151; PMCID: PMC6058046.
5. The figures in the supplemental file regarding how the authors utilized methods of Sun et al 2020 and Szabo et al., 2019 can be removed, thus the methods can merely be described and cited. This suggestion to remove the content is at the discretion of the editor.
6. Lastly protein restriction largely induces FGF21, it is largely known that FGF21 is a metabolic hepetokine that improve measurement of metabolic health, it would be of great value for the authors to include FGF21.
See citation below
Laeger T, Henagan TM, Albarado DC, Redman LM, Bray GA, Noland RC, Münzberg H, Hutson SM, Gettys TW, Schwartz MW, Morrison CD. FGF21 is an endocrine signal of protein restriction. J Clin Invest. 2014 Sep;124(9):3913-22. doi: 10.1172/JCI74915. Epub 2014 Aug 18. PMID: 25133427; PMCID: PMC4153701.
https://pubmed.ncbi.nlm.nih.gov/?term=FGF21+protein+restriction&sort=date
Author Response
Response to reviewers
Dear Reviewer,
Please find the reply for your suggestions and comments.
Open Review
( ) I would not like to sign my review report
(x) I would like to sign my review report
English language and style
( ) Extensive editing of English language and style required
( ) Moderate English changes required
( ) English language and style are fine/minor spell check required
(x) I don’t feel qualified to judge about the English language and style
|
Yes |
Can be improved |
Must be improved |
Not applicable |
|
|
Does the introduction provide sufficient background and include all relevant references? |
( ) |
(x) |
( ) |
( ) |
|
Are all the cited references relevant to the research? |
(x) |
( ) |
( ) |
( ) |
|
Is the research design appropriate? |
(x) |
( ) |
( ) |
( ) |
|
Are the methods adequately described? |
( ) |
(x) |
( ) |
( ) |
|
Are the results clearly presented? |
(x) |
( ) |
( ) |
( ) |
|
Are the conclusions supported by the results? |
(x) |
( ) |
( ) |
( ) |
Comments and Suggestions for Authors
Manuscript Number: nutrients-1751807
Dietary Protein Restriction Reverses Metabolic Dysfunction in Patients with Metabolic Syndrome in a Randomized, Controlled Trial
The authors of the original data review manuscript titled “Dietary Protein Restriction Reverses Metabolic Dysfunction in Patients with Metabolic Syndrome in a Randomized, Controlled Trial” confronts an important question regarding the effects of PR versus CR in humans. While many studies in rodents have identified the effects of PR on metabolic health and lifespan in both lean and obese conditions, this study highlights the improvements made during PR. The PR dietary interventions are largely supported by feeling less hungry as observed during CR, though CR is established as the gold standard to improve health. The data from this manuscript aligns with many studies to improve the translational aspects of PR on health as well in specific tissues, namely adipose tissue and the gut.
Overall, this manuscript is well written, providing new data regarding the effects of PR during diabetes with obesity.
Minor Concerns:
1. The title uses “reverses” in the context as to cure metabolic dysfunction, while the effects of
metabolic dysfunction are controlled from the stated metabolic endpoints only during dietary
interventions. The authors may want to consider something along the lines
- Dietary Protein Restriction Improves Metabolic Health in 2 Patients with Metabolic Syndrome in a Randomized, Controlled Trial.
- Dietary Protein Restriction Improves Metabolic Endpoints in 2 Patients with Metabolic Syndrome in a Randomized, Controlled Trial.
- Measures of Metabolic Health are Improved during Dietary Protein Restriction……
- Dietary Protein Restriction Improves Metabolic Endpoints in 2 Patients with Metabolic Syndrome in a Randomized, Controlled Trial.
This reviewer makes this suggestion due to Suppl Fig 4, where all improvements return to basal line.
We appreciate the title suggestions provided by the reviewer. The title was changed to “Dietary Protein Restriction Improves Metabolic Dysfunction in Patients with Metabolic Syndrome in a Randomized, Controlled Trial.”
The reviewer suggests that patient outcomes assessed at follow-up return to baseline. However, the results shown in Sup. Fig 4 demonstrates that there were no significant changes in the parameters evaluated between the last week of treatment (PR or CR) and 1 month after treatment. The Sup. Fig 4 demonstrates that the metabolic improvements promoted by PR or CR are maintained up to 1 month after treatment. A more careful analysis reveals that the patients evaluated in the follow-up did not return to the baseline values demonstrated before submitting to the PR or CR protocol. This information was clarified in the text (lines 373-374)
Abstract conclusion statement: The conclusion statement presents a pretentious undertone PR compared to CR “the isocaloric characteristic of the PR intervention makes this approach a more attractive and less drastic dietary strategy in clinical settings and has more significant potential to be used as adjuvant therapy for people with metabolic syndrome”.
We understand the reviewer’s concern about this statement. However, the results carefully demonstrated in this controlled clinical trial allow us to suggest that PR has the potential to be used as a dietary strategy in the control of metabolic syndrome. More importantly, we show that this is a safe approach, and in the text, we added the limitations and need of further study to confirm our findings.
Major Concerns
3. Figure 2: The authors used an ANOVA for the statistical reference. Given that the body weights are different after the intervention the appropriate stat reference should be an ANCOVA, this will consider the variations and changes in BW before and after the dietary interventions for each subject. Though Figure 3 in the supplemental seems to be more accountable for changes seen during the intervention by plotting value -X by corresponding BW or changes in energy ---see publication Tim Hollstein, Alessio Basolo, Takafumi Ando, Jonathan Krakoff, Paolo Piaggi, Reduced adaptive thermogenesis during acute protein-imbalanced overfeeding is a metabolic hallmark of the human thrifty phenotype, The American Journal of Clinical Nutrition, Volume 114, Issue 4, October 2021, Pages 1396–1407, https://doi.org/10.1093/ajcn/nqab209
We appreciate the reviewer’s suggestion. The decision to use the statistical method ANOVA mixed-design analysis of variance was carefully discussed with a statistical core, which guided us to use it, especially for our data. (https://www.rti.org/rti-press-publication/mixed-model-approach-intent-treat; https://psych.wisc.edu/Brauer/BrauerLab/wp-content/uploads/2014/04/Murrar-Brauer-2018-MM-ANOVA.pdf).
- Gut microbiota. The authors state that there is no changes in the microbiome during short-term PR, yet make no mention of the CR diets or having what is consider NP (normal protein). Please discuss when interpreting your results
- Can the authors expound on the differences of CR and PR… see
1. Martin A, Ecklu-Mensah G, Ha CWY, Hendrick G, Layman DK, Gilbert J, Devkota S. Gut microbiota mediate the FGF21 adaptive stress response to chronic dietary protein-restriction in mice. Nat Commun. 2021 Jun 22;12(1):3838. doi: 10.1038/s41467-021-24074-z. PMID: 34158480; PMCID: PMC8219803. - Dong TS, Luu K, Lagishetty V, Sedighian F, Woo SL, Dreskin BW, Katzka W, Chang C, Zhou Y, Arias-Jayo N, Yang J, Ahdoot A, Li Z, Pisegna JR, Jacobs JP. A High Protein Calorie Restriction Diet Alters the Gut Microbiome in Obesity. Nutrients. 2020 Oct 21;12(10):3221. doi: 10.3390/nu12103221. PMID: 33096810; PMCID: PMC7590138.
- Chen X, Song P, Fan P, He T, Jacobs D, Levesque CL, Johnston LJ, Ji L, Ma N, Chen Y, Zhang J, Zhao J, Ma X. Moderate Dietary Protein Restriction Optimized Gut Microbiota and Mucosal Barrier in Growing Pig Model. Front Cell Infect Microbiol. 2018 Jul 18;8:246. doi: 10.3389/fcimb.2018.00246. PMID: 30073151; PMCID: PMC6058046.
Regarding the microbiota analysis in this study, we only had access to samples from individuals submitted to PR. We did not evaluate patients undergoing CR. This was a limitation in our study, and we entered this information in the limitations section (lines 542-543). Therefore, we did not develop a discussion comparing the microbiota changes between CR and PR. However, we report our findings on PR and discuss them based on clinical studies cited in the references 40,41,42.
Paper 1: In this article, the main finding is that the compositional changes in the intestinal microbiota were mainly related to a protein-restricted diet supplemented with fiber (Cellulose). In our study, the PR was 10% protein compared to the amount of an ideal diet of 20% protein without fiber supplementation.
Paper 2: The article reports that the subjects underwent HPD (30% protein, 40% carbohydrates and 30% fat) and NPD (15% protein, 55% carbohydrates and 30% fat). In our study, individuals undergoing PR had a diet (10% protein, 60% carbohydrates and 30% fat). Here we have a clear difference between the PR of our study compared to the NPD of the evaluated study. The conclusion of this study reinforces our findings, demonstrating that microbial changes are related to high-protein dietary interventions.
Paper 3: in this article, they used 3 types of diets with protein restrictions 12%, 15%, and 18%. They reported that moderate protein restriction (15%) is able to significantly alter the composition of microbial populations and epithelial cell proliferation. In our study, the PR was 10% of protein restriction. This paper also reinforces our findings that protein restriction lower than 15% is not enough to significantly alter the intestinal microbiota composition.
- The figures in the supplemental file regarding how the authors utilized methods of Sun et al 2020 and Szabo et al., 2019 can be removed, thus the methods can merely be described and cited. This suggestion to remove the content is at the discretion of the editor.
We appreciate the reviewer’s comment. We inserted the information from the articles (Szabo et al., 2019 and Sun et al., 2020) in the legends of the supplementary figures related to the deconvolution analysis to facilitate the interpretation of our results and show the reader where the analyzed results come from. However, if the editor requests it, we can remove them.
- Lastly protein restriction largely induces FGF21, it is largely known that FGF21 is a metabolic hepetokine that improve measurement of metabolic health, it would be of great value for the authors to include FGF21.
See citation below
Laeger T, Henagan TM, Albarado DC, Redman LM, Bray GA, Noland RC, Münzberg H, Hutson SM, Gettys TW, Schwartz MW, Morrison CD. FGF21 is an endocrine signal of protein restriction. J Clin Invest. 2014 Sep;124(9):3913-22. doi: 10.1172/JCI74915. Epub 2014 Aug 18. PMID: 25133427; PMCID: PMC4153701.
https://pubmed.ncbi.nlm.nih.gov/?term=FGF21+protein+restriction&sort=date
We appreciate this interesting suggestion from the reviewer. However, we do not have enough plasma samples to perform ELISA evaluating FGF21 in our studied subjects. We will plan on this evaluation for further studies.
Reviewer 2 Report
Although the study sample is quite small, which is the main limitation of this research, the methodology is of high quality. Given the challenges in conducting clinical nutrition research the main strenghts of the present study are the facts that patients were hospitalized and adherence to the diet was assessed daily after each meal by a nutritionist. The strenghts and limitations of the study should be emphasized in the separate section.
Author Response
Response to reviewers
Dear Reviewer,
Please find the reply to your suggestions.
Open Review
( ) I would not like to sign my review report
(x) I would like to sign my review report
English language and style
( ) Extensive editing of English language and style required
( ) Moderate English changes required
(x) English language and style are fine/minor spell check required
( ) I don’t feel qualified to judge about the English language and style
|
Yes |
Can be improved |
Must be improved |
Not applicable |
|
|
Does the introduction provide sufficient background and include all relevant references? |
(x) |
( ) |
( ) |
( ) |
|
Are all the cited references relevant to the research? |
(x) |
( ) |
( ) |
( ) |
|
Is the research design appropriate? |
(x) |
( ) |
( ) |
( ) |
|
Are the methods adequately described? |
(x) |
( ) |
( ) |
( ) |
|
Are the results clearly presented? |
(x) |
( ) |
( ) |
( ) |
|
Are the conclusions supported by the results? |
(x) |
( ) |
( ) |
( ) |
Comments and Suggestions for Authors
Although the study sample is quite small, which is the main limitation of this research, the methodology is of high quality. Given the challenges in conducting clinical nutrition research the main strenghts of the present study are the facts that patients were hospitalized and adherence to the diet was assessed daily after each meal by a nutritionist. The strenghts and limitations of the study should be emphasized in the separate section.
We appreciate the reviewer’s comments and suggestions.
Following the suggestions reported here, we included a paragraph in the discussion aborting the main limitations of our study (lines 537-546).
Reviewer 3 Report
I congratulate the authors for the huge work done in this clinical trial.
Here are my suggestions and correction:
1- Please change the title, your study is not on protein restriction, but use a correct protein distribution, use another title e.g Proper protein intake reverses metabolic dysfunction in patients with...
2- Lines 40-41 The sentence is not clear.
3-Lines 57-59 Otherwise CR contributes to reshaping lipoproteins and decreases levels of proinflammatory cytokines. Please read this paper 'Anti-inflammatory effects of diet and caloric restriction in metabolic syndrome' link to publication https://rdcu.be/cN8MY.
4-Lines 67-68, This sentence is not supported by reference 13. In the reference who cited the results clearly indicate that only higher animal-to-plant protein ratio (extreme-quartile HR = 1.23; 95% CI: 1.02, 1.49; P-trend = 0.01) and higher meat intake (extreme-quartile HR = 1.23; 95% CI: 1.04, 1.47; P-trend = 0.01) were associated with increased mortality. Intakes of fish, eggs, dairy, or plant protein sources were not associated with mortality.
5-Experimental Design. Why is the study not registered on clinicaltrial.gov?
6-Lines 88-89, after the end of the sentence, insert as described below, because the strict inpatient environment is not immediately clear to the readers.
7-Lines 111-112 Please insert the methods for the determination of energy expenditure in the subjects here.
8-Lines 116 the protein 0.8 g of protein/kg body weight is not a protein restriction but is a proper protein intake. 'In adults, the recommended daily amount of protein ranges from 0.80 to 0.83 g per kilogram of body weight for both men and women with modest levels of physical activity. Recommended amounts for children and pregnant or lactating women are higher, to allow for the deposition of body tissues and the secretion of milk.'
link https://knowledge4policy.ec.europa.eu/health-promotion-knowledge-gateway/dietary-protein_en#:~:text=In%20adults%2C%20the%20recommended%20daily,and%20the%20secretion%20of%20milk.
9-line 130, please indicate the model of indirect calorimetry you used.
10-Lines 158-159 the lean mass is not the same as free-fat mass (FFM). FFM is calculated with BIA, whether the lean mass is calculated with the use of DXA. So, please correct in the entire manuscript the lean mass with free-fat mass (FFM).
11-Line 233, please insert the description of the mean of weight and BMI.
12-Table 1 in the section characteristic insert the weight not only the height.
Table 1 is not clear, is better to split in 2 tables:
Table 1 with characteristic, primary and secondary outcomes;
Table 2 with variables before and after, please insert which of the two is CR and PR.
13-line 251, not lean mass but FFM.
14-Section 3.5 the analysis of the results, in particular, the HOMA-beta and HOMA-IR could be flawed by a single sample even in CR that in PR group, the two samples are present in the upper part of the two graphs. These 2 samples are probably outliers that have a great impact on the analysis, please reanalyze these data without these two samples.
15-Line 369, insert a brief explanation of FDR.
16-Change the discussion about HOMA in light of the new analysis without outliers.
Author Response
Response to reviewer:
Dear Reviewer,
Please find the reply to your comments and suggestions.
Open Review
( ) I would not like to sign my review report
(x) I would like to sign my review report
English language and style
( ) Extensive editing of English language and style required
( ) Moderate English changes required
(x) English language and style are fine/minor spell check required
( ) I don’t feel qualified to judge about the English language and style
|
Yes |
Can be improved |
Must be improved |
Not applicable |
|
|
Does the introduction provide sufficient background and include all relevant references? |
( ) |
(x) |
( ) |
( ) |
|
Are all the cited references relevant to the research? |
( ) |
(x) |
( ) |
( ) |
|
Is the research design appropriate? |
( ) |
(x) |
( ) |
( ) |
|
Are the methods adequately described? |
( ) |
( ) |
(x) |
( ) |
|
Are the results clearly presented? |
( ) |
( ) |
(x) |
( ) |
|
Are the conclusions supported by the results? |
( ) |
( ) |
(x) |
( ) |
Comments and Suggestions for Authors
I congratulate the authors for the huge work done in this clinical trial.
Here are my suggestions and correction:
1- Please change the title, your study is not on protein restriction, but use a correct protein distribution, use another title e.g Proper protein intake reverses metabolic dysfunction in patients with...
We are grateful for the suggestion. We chose this title because of the restrictive character compared to the standard western diet (PR 0.8 protein/kg body weight vs 1.5 g protein/kg body weight in the western diet). In addition to 0.8 protein/kg of body weight, the minimum recommended for a diet to be safe for the wide population and the reduction was an ethical concern. Now that we show that acute reduction is safe from a clinical perspective, we may be allowed to investigate the effect of lower levels of protein in further studies. Also, we reviewed the title, now it is “Dietary Protein Restriction Improves Metabolic Dysfunction in Patients with Metabolic Syndrome in a Randomized, Controlled Trial.”. Please see if you think this is more suitable.
2- Lines 40-41 The sentence is not clear.
We changed the sentence in the abstract to make it clearer to the reader.
3-Lines 57-59 Otherwise CR contributes to reshaping lipoproteins and decreases levels of proinflammatory cytokines. Please read this paper ‘Anti-inflammatory effects of diet and caloric restriction in metabolic syndrome’ link to publication https://rdcu.be/cN8MY.
We are grateful for the suggestion and insert the information on pro-inflammatory cytokines and their respective reference. (Lines 58-59).
4-Lines 67-68, This sentence is not supported by reference 13. In the reference who cited the results clearly indicate that only higher animal-to-plant protein ratio (extreme-quartile HR = 1.23; 95% CI: 1.02, 1.49; P-trend = 0.01) and higher meat intake (extreme-quartile HR = 1.23; 95% CI: 1.04, 1.47; P-trend = 0.01) were associated with increased mortality. Intakes of fish, eggs, dairy, or plant protein sources were not associated with mortality.
We appreciate the reviewer’s attention. We removed the wrong citation and inserted a new, more appropriate reference.
5-Experimental Design. Why is the study not registered on clinicaltrial.gov?
We have registered our study at ensaiosclinicos.gov.br because it is a register of responsibility and quality, and also because it is recognized by the WHO. (https://www.who.int/clinical-trials-registry-platform/network/primary-registries/rebec-(registro-brasileiro-de-ensaios-clinicos).
6-Lines 88-89, after the end of the sentence, insert as described below, because the strict inpatient environment is not immediately clear to the readers.
We appreciate the suggestion. We corrected this sentence as directed by the reviewer.
7-Lines 111-112 Please insert the methods for the determination of energy expenditure in the subjects here.
We appreciate the suggestion. We insert the method used to determine the individuals’ basal energy expenditure.
8-Lines 116 the protein 0.8 g of protein/kg body weight is not a protein restriction but is a proper protein intake. ‘In adults, the recommended daily amount of protein ranges from 0.80 to 0.83 g per kilogram of body weight for both men and women with modest levels of physical activity. Recommended amounts for children and pregnant or lactating women are higher, to allow for the deposition of body tissues and the secretion of milk.’
link https://knowledge4policy.ec.europa.eu/health-promotion-knowledge-gateway/dietary-protein_en#:~:text=In%20adults%2C%20the%20recommended%20daily,and%20the%20secretion%20of%20milk.
We standardized the protein restriction protocol at 0.8g of protein/kg body weight for 2 reasons:
- Because it is a clinical trial using patients with metabolic syndrome, we did not have the permission of the human ethics committee to use a diet with protein restriction below the minimum stipulated by the nutritional guide, in this case 0.8g.
- One of the objectives of this study is to provide an effective and affordable/applicable dietary intervention for a non-hospital environment. Thus, we stipulated the protein restriction as the minimum allowed 0.8g/kg of protein/kg body weight.
9-line 130, please indicate the model of indirect calorimetry you used.
We insert the information from the indirect calorimetry model. (line 131-132)
10-Lines 158-159 the lean mass is not the same as free-fat mass (FFM). FFM is calculated with BIA, whether the lean mass is calculated with the use of DXA. So, please correct in the entire manuscript the lean mass with free-fat mass (FFM).
We’ve replaced all lean mass terms swapping for free-fat mass.
11-Line 233, please insert the description of the mean of weight and BMI.
The description of the mean weight and BMI are reported in the lines 258-260
12-Table 1 in the section characteristic insert the weight not only the height.
We did not enter body weight information in the table because this information was shown graphically in Figure 1A. We seek to avoid redundancy.
Table 1 is not clear, is better to split in 2 tables:
Table 1 with characteristic, primary and secondary outcomes;
Table 2 with variables before and after, please insert which of the two is CR and PR.
Thank you for the suggestion the table was split.
13-line 251, not lean mass but FFM.
We made the correction. Line 265
14-Section 3.5 the analysis of the results, in particular, the HOMA-beta and HOMA-IR could be flawed by a single sample even in CR that in PR group, the two samples are present in the upper part of the two graphs. These 2 samples are probably outliers that have a great impact on the analysis, please reanalyze these data without these two samples.
We are grateful for the reviewer’s observation. We have redone the analyzes excluding 2 two individual outliers from both CR and PR. Even with the exclusion of the 2 CR outliers, HOMA-B and HOMA-IR present a significant difference p=0.0132, p=0.0090, respectively.
When we exclude those from the 2 PR outliers, the HOMA-B still shows a significant difference p=0.0025. However, the HOMA-IR no longer shows a significant difference p=0.0835. We addressed this new information in the results and discussion.
15-Line 369, insert a brief explanation of FDR.
We made the correction.
16-Change the discussion about HOMA in light of the new analysis without outliers.
We include in the results (Lines 359-364) and discussion (Lines 479-484) the results referring to the new analysis of the HOMA-IR in the individuals who underwent PR without the outliers as explained above.
Round 2
Reviewer 1 Report
Thank you for your attention to the comments of this reviewer.
Author Response
We are grateful for this reviewer's comments and suggestions.
Reviewer 3 Report
The authors have provided the corrections and suggestions of the reviewer, implementing the method and correcting the results, even if I do not see the relevant change in graphs of HOMA-beta of CR and PR. Moreover, the term protein restriction is not proper correctly, but the author explains the reason for the use of this term. The other issues of this paper were resolved, so only this minor correction should be done.
Author Response
We appreciate the reviewer's comments and suggestions. We created a supplementary figure (Supplementary Figure 4) showing the analysis of HOMA-B and HOMA-IR with the removal of the 2 outliers from both protocols. This information is described in the results (line 365).
We chose to create the supplementary figure because the results of improving insulin sensitivity are supported by the hyperinsulinemic-euglycemic clamp, which is recognized as the gold standard for this type of assessment. Another reason we chose to keep the raw data from HOMA-B and HOMA-IR is because we would like to show the effects of both treatments in all patients evaluated in our study. However, this reviewer's concern is reported in the results section and discussed in the discussion section. We also updated the sequences of the supplementary figures (Lines 556-560).